# AI Based Digital Twin Model for Cattle Caring

**DOI:** 10.3390/s22197118

**Published:** 2022-09-20

**Authors:** Xue Han, Zihuai Lin, Cameron Clark, Branka Vucetic, Sabrina Lomax

**Affiliations:** 1Centre of IoT and Telecommunication (CIoTT), School of Electrical and Information Engineering, Faculty of Engineering, University of Sydney, Camperdown, NSW 2006, Australia; 2Livestock Production and Welfare Group, School of Life and Environmental Sciences, Faculty of Science, University of Sydney, Camden, NSW 2570, Australia

**Keywords:** digital twin, AI, deep learning, LSTM model

## Abstract

In this paper, we develop innovative digital twins of cattle status that are powered by artificial intelligence (AI). The work is built on a farm IoT system that remotely monitors and tracks the state of cattle. A digital twin model of cattle based on Deep Learning (DL) is generated using the sensor data acquired from the farm IoT system. The physiological cycle of cattle can be monitored in real time, and the state of the next physiological cycle of cattle can be anticipated using this model. The basis of this work is the vast amount of data that is required to validate the legitimacy of the digital twins model. In terms of behavioural state, this digital twin model has high accuracy, and the loss error of training reach about 0.580 and the loss error of predicting the next behaviour state of cattle is about 5.197 after optimization. The digital twins model developed in this work can be used to forecast the cattle’s future time budget.

## 1. Introduction

Digital twins are virtual digital representations of physical objects, in which the physical object and its corresponding virtual digital representation interact remotely in real time [1]. A digital twin model incorporates multi-disciplinary, multi-physical quantity, multi-scale, and multi-probability simulation processes and fully utilises physical models, sensor updates, operation histories, and other data [2]. In addition, digital twins complete the mapping in virtual space so that the full life cycle process of associated entity equipment is reflected [3]. Digital twins are a transcendental idea that can be regarded as one or more crucial and interdependent digital mapping systems for the actual object [4,5].

Connectivity, modularity, and autonomy between virtual and actual items can all be realised with digital twins. It can be accomplished across the whole production process from product design through product system engineering to production planning, implementation and intelligence, resulting in a self-optimizing closed loop [6]. To put it another way, by connecting the actual object with the virtual number, the real object may offer real information to optimise the digital model, and the digital model can foresee potential situations to alter the real object. The two complement each other to create a self-closing optimisation mechanism [7]. Nowadays, digital twins have been increasingly employed in a variety of industries, including product design, product manufacturing, medical analysis, engineering construction and other areas [8]. As a result, digital twins can be seen as a major force behind the intelligent manufacturing paradigm [9]. Digital twins have recently been deployed in a variety of fields, including livestock farming [10,11].

Deep learning (DL) is a new direction in machine learning that is being introduced to bring it closer to the goal of AI and has made tremendous progress in solving issues that were previously unsolvable in AI. It has proven to be so effective in detecting complicated structures in high-dimensional data that it might be used in a wide range of scientific, business and government applications [12,13,14]. The long short-term memory network (LSTM) is a type of cyclic neural network and one of the deep learning algorithms that can analyse and forecast critical time with very long intervals and delays in time series [15,16]. In a long time series, the LSTM neural network algorithm can determine which information should be stored and which should be discarded [17]. The development of digital twins relies heavily on accurate time series prediction. Internal and external disruptions might result in time series that are exceedingly nonlinear and random. Complex object time series prediction may be employed at any stage of their life cycle, which is also a major component of the digital twin model [18,19]. Therefore, it is extremely dependable to use the LSTM model to build digital twins.

This research is primarily focused on the direction of intelligent livestock monitoring in the agricultural environment. For a long time, Australia has been a major producer of animal husbandry, and milk and beef production and export have also been at the forefront of the global [20]. Cattle statuses may be monitored in real time to enable breeders better determine their cattle’s health and enhance meat and milk output correspondingly [21]. As a result, agriculture’s evolution toward intelligence is a critical stage of growth [22]. This project aims to create a digital twin model for each individual bovine, which will allow for improved monitoring of cattle status at the digital and information levels, as well as the advancement of Australian animal husbandry. The main contribution of this study is that it developed an intelligent digital twins approach using an LSTM neural network to give a range of behavioural detection and prediction of cattle’s state, such as impending physiological cycles, among other things. The digital twins model is significantly based on massive volumes of data reflecting cattle location, movement and free grazing time, etc, collected by the farm IoT monitoring system. The digital twin model has some limitations; for example, when the amount of sampled data is inadequate, the model’s accuracy is unsatisfactory. As a result, this model needs a considerable amount of sample data.

The outline of the paper is given below. First in Section 2, the current related work of digital twin is summarized. After that, in Section 3, necessary data mining and data analysing for the IoT system are carried out. In this part, most of the data processing work is accomplished with the help of MATLAB. In Section 4, cattle’s behaviour states are modelled by training the LSTM neural network in the digital twin model and cattle’s states in the next cycle are predicted by using this deep learning technique. In Section 5, The accuracy of the trained LSTM model is discussed and verified. Finally, Section 6 deduces a proper conclusion.

## 2. Related Work

The concept of digital twins can be applied in many areas. For example, for wind power plants, cloud-based technology integrates technological and commercial data into a single digital twin through augmented reality (AR) and applies to multiple wind power plants to realize real-time monitoring of power plants [3]. In manufacturing, Schleich et al. [2] propose a conceptual integrated reference model for design and manufacturing that provides the first theoretical framework for digital twins in industrial applications [2]. In addition, digital twins are also used for product prediction and health management. This method effectively utilizes the interaction mechanism and fusion data of digital twins [8].

In the field of agriculture, more and more farmers are committed to the establishment of intelligent farms. The concept of intelligent agriculture mainly includes sensors, tracking systems, innovative digital technologies, data analysis and so on. The application of modern digital technology to farms can improve the efficiency of farm management [9,10]. More specifically, Yang et al. [7] come up with a digital farm management system that can effectively track production. In particular, this system uses smartphones to collect data, this is an efficient solution for precise vegetable farm management [7]

However, digital twins are rarely used on farms, there are only a few isolated cases. Digital twins are already being used in innovative internet-based applications, and digital twins can influence farm management [4]. A digital disease management system for dairy cows has been established, which realizes the digital management of dairy cows, systematic management of basic information of dairy cows, health assessment, electronic medical records and disease prevention. This system can effectively manage the disease of cows on the dairy farm [23]. In [13], Wagner et al. [13] use machine learning to detect the health of cows and predict when they would behave. They use different algorithms in machine learning to predict the activity duration of cows, including the K-neighborhood algorithm, the LSTM algorithm, the H-24 algorithm, and so on. The K-nearest neighbour algorithm performs the best after analysis and comparison. However, their study needs to be based on a much larger data set and needs to take into account the circadian nature of activity rhythms.

In addition, the application of an LSTM neural network to the establishment of the digital twin model used on a farm is also rare, but it’s been used in many other ways and has been very successful. Hu et al. [18] propose a hybrid time series prediction model based on global empirical mode decomposition, LSTM neural network and Bayesian optimization, and apply it to the establishment of the digital twin model. They use their digital twin models to predict wind speeds in wind turbines and wave heights in ocean structures. The results show that the proposed model can obtain accurate time series prediction [18]

Although the application of digital twins in farm management is still in the early stage of development, it is not impossible to establish digital twins for each cow on this bovine disease digital management system. With the establishment of digital twins, the cattle farm can become an autonomous, adaptive system in which intelligent digital twins can operate, decide and even learn without human on-site or remote intervention [4].

## 3. Data Mining and Analysing

This section primarily discusses the processing method of the data sets, i.e., the original data measured by sensors of the farm’s IoT system. This data set is systematically treated in preparation for future use of the modelling. Particularly, the data sets of the cattle’s states are analysed, and a digital twin model of the cattle is produced using these data sets. A vast amount of data may be used to evaluate the model’s correctness, and the state of the cattle can then be predicted.

### 3.1. Data Processing

The raw data set for the sensor contains 98 cattle of various breeds and genders. There are eight categories used to classify cattle’s status: Resting, Rumination, High Activity, Medium Activity, Panting (Heavy Breathing), Grazing and Walking. Detailed descriptions for different cattle states are shown in Table 1. Each sensor takes a minute-by-minute reading of the cows’ real-time status, with each cow having 74,455 data points collected between AU_time 8:06 a.m. on August 10 and 1:01 a.m. on 1 October 2019. Five cattle breeds are represented in the data sets: Angus, Brahman, Brangus, Charolais and Crossbred. This section focuses on the systematic processing of these data, including data segmentation, data cleaning, and data calculating.

#### 3.1.1. Data Segmentation

The first step in data processing is the segmentation. The data are grouped by cattle of the same sex and breed. Because the original data are massive, we segment the data using RStudio and R programming language. Table 2 shows the number of cows segmented and integrated. A vast amount of data facilitates the analysis of overall data characteristics and avoid errors caused by individual and particular data. As a result, the resting state of Brahman’s Female is used to demonstrate data processing and prediction.

#### 3.1.2. Data Cleaning

When the sensor detects and transmits the status of the cow, it also sends a lot of invalid data. The accuracy of the original data will be considerably influenced if using these data directly. Therefore, the initial step is to clear up the corrupted data.

Because the data returned by the sensor represents the cattle’s states at a specific point in time, quantifying that state is critical for further design. In this work, the time of various states each hour in minutes is taken as the research object. Because corrupted or invalid data usually aggregate, identifying the point at which incorrect data arrives as 0 is not precise. For example, if there is a large amount of damage data in an hour, the rest state of the cattle for that hour will be marked as 0 min, which will affect the calculation of the average single period. Therefore, deleting corrupted data and the corresponding time serial number, to ensure that they are not included in the calculation of the average period.

The flow chart of data cleaning is shown in Figure 1. Data cleaning mainly focuses on the segmented data to clean and organize and finally obtains the cleaned data and its corresponding time series. This step primarily calculates the resting time of cattle in each hour. If it exists any corrupted data during the calculated hour, that hour’s data will be destroyed.

### 3.2. The State of Cattle throughout the Sampling Period

Acquiring the cattle’s state changes across the sample period needs to average one group’s data of cows due to large amount of discrete and lost data from a single cow. For example, averaging the resting time per hour of 14 Brahman females can determine variations in the resting state of Brahman treated during the sample period. The time series after data cleaning are different between each cattle’s data set, since invalid data collected by sensors in the farm’s IoT system is usually a random process. Therefore, the data processing in this step is to average the state data of the cattle with the same time serial number and obtain the state curve of the cattle in the whole cycle. The process flow chart of an average state time for several cattle can be found in Figure 2.

The state diagram of cattle in the entire cycle can be obtained after the program has been executed. Figure 3 shows the calculated hourly rest time of the cattle in the whole cycle (Brahman Female). The number on the abscissa corresponds to the corresponding day, which includes all 24 h. The ordinate represents the rest time corresponding to this hour in minutes.

Figure 4 is a detailed zoomed-in part of Figure 3 and located between days 16 and 20. It is obvious that the rest time of cattle varies periodically with a cycle of one day. The peaks of the daily rest time can be found in both early morning and late-night while the valleys can usually be identified at forenoon and afternoon hours.

### 3.3. The Average 24 h State of Cattle

The averaged single rest cycle data result (which is 24 h) of a single cattle is plotted in Figure 5. The entire sampling cycle is approximately 52 days as shown in Figure 3. The abscissa refers to the o’clock, i.e., from 0:00 to 23:59, and the ordinate relates to the rest period in minutes at this hour (Brahman Female). The average period’s plot is flatter than a single period’s plot. However, the trend and structure of these two are nearly identical, and a single cycle has more individual points and noises.

### 3.4. Fitting Curve for the Average State Period (24 h)

Curve fitting is commonly used to obtain the data relationship for such irregular curves. Typical fitting methods include minimum binomial fitting, exponential function fitting, power function fitting, and hyperbola fitting. Different fitting approaches are compared in this section to obtain the most ideal mathematical model [24,25].

Four fitting approaches are utilized to fit the 24-h average rest duration of cattle: Gaussian fitting, Sum of Sine fitting, Polynomial fitting, and Fourier fitting. The independent variable is the time, and the dependent variable is the rest period of cattle corresponding to that time while fitting the curve. The relationship between the time and the associated rest time can be established, and the curve of the cattle’s rest period throughout the day can be obtained. As indicated in Table 3, Gaussian (item number 8) fitting is found to be the most accurate model among all candidates in terms of the fitting variance result. The error variance of Gaussian fitting is only 3.0037, which is much smaller than that of other fitting methods. The fitted curve shape is depicted in Figure 6, it is basically consistent with that of the average period in Figure 5.

The formula of the fitting curve (Gauss eight-term) formula is:(1)$$\begin{array}{cc} f(x)= & 51.29e^{{\left(-\frac{x-2.823}{2.957}\right)}^{2}}+44.42e^{{\left(-\frac{x-24.19}{3.936}\right)}^{2}}+1.378\times 10^{14}e^{{\left(-\frac{x+40.24}{7.546}\right)}^{2}}+19.29e^{{\left(-\frac{x-13.55}{3.22}\right)}^{2}}\\ \; & +16.18e^{{\left(-\frac{x-19.06}{0.9367}\right)}^{2}}+19.25e^{{\left(-\frac{x-4.588}{0.5802}\right)}^{2}}+29.29e^{{\left(-\frac{x-20.39}{1.802}\right)}^{2}}+20.45e^{{\left(-\frac{x-9.812}{2.834}\right)}^{2}} \end{array}$$

In Equation (Equation 1), *x* is the clock of a day, while f(x) denotes the rest time within one hour of that clock. Regarding the low standard deviation and variance of this fitting result, this model is considered to be the proper candidate to describe the resting time of cattle in a day for Brahman Females. The models for other breeds, genders and states can be obtained in the same way.

### 3.5. Noise Reduction Using Low-Pass Finite Impulse Response (FIR) Filter

Throughout the sample period, the cattle’s condition varies on daily basis. The plot of the entire activity cycle contains noise and outliers in Figure 3. Therefore, denoising the sampled data is required.

FIR andInfinite Impulse Response (IIR) are two types of digital filters that are extensively employed. In theory, an IIR function’s filtering effect is superior to that of an FIR function of the same order, but divergence can occur. The IIR digital filter has a high precision for amplitude-frequency characteristics, and with a non-linear phase, it is suited for audio signals that are insensitive to phase information. FIR digital filters have lesser amplitude-frequency precision than IIR digital filters. However, the phase is linear, meaning the time difference between signals of various frequency components remains unaltered after going through the FIR filter. In addition, the calculation time delay is relatively tiny, it is suited for real-time signal processing [26,26]. Because the state of the cattle is time-series data, it is critical to ensure that the filtered phase remains constant. Therefore, in this work, we use the FIR low-pass filter for denoising.

Cattle monitoring data are sampled once every 60 s in this study, resulting in a sampling frequency of around 0.0167 Hz. This is a low-frequency sampling signal, and the noise is present between each sampling. Noise frequency is more extensive than sampling frequency, so the signal between 0 and 0.0167 Hz is kept while the signal above 0.0167 Hz is eliminated. In Figure 7, the filter length is set to 5, and the filter’s shape corresponds to its frequency. The filtered result is depicted in Figure 8, which uses the resting time of a Brahman Female’s cow as an example.

Figure 8b is a local detailed version of Figure 8a, focusing on the comparison of before using FIR filtering and after using FIR filtering from the 16th to the 20th day. Data performance is optimized after the introduction of the FIR filter for smooth signal processing, and the data trend can be clearly identified.

After going through the FIR filter, Figure 9 provides an image of a single rest period (one day, Day 17). In comparison to Figure 5, it exhibits the same trend, i.e., one day’s rest time after filtering is nearly the same as one day’s typical rest time. This feature demonstrates that the cattle’s condition changes on a regular basis. It also indicates that the FIR filtered signal is effective and precise. The FIR filter effectively minimizes noise and eliminates outliers and gross inaccuracy. As a result, the signal filtered by the FIR filter can be used for subsequent modelling and prediction.

## 4. Prediction Based on LSTM Model

In DL, the LSTM network is a unique RNN model. Its unique structural design allows it to avoid long-term reliance. The default nature of LSTM is to remember information from a long time ago [12,17,27,28]. In this section, we employ the LSTM model to forecast the status of cattle based on the above research content. To be more explicit, the structure and properties of LSTM and how to construct an LSTM model are first discussed. Second, using the LSTM model, the cattle status is modelled and forecasted. Finally, the model is optimized in order to improve its accuracy.

### 4.1. Build the LSTM Model of the Cattle State

The program flow chart for establishing the LSTM model is shown in Figure 10. First, import the data previously filtered by the FIR filter, and divide it into a test set and a training set. Second, the LSTM model is created. Setting parameters: the number of input neurons, output neurons, hidden neurons, learning rate, batch size, epoch size (i.e., the number of training cycles) and the number of LSTM layers [29,30]. The loss error is chosen as the mean square error, and the LSTM neural network is trained using the Adam optimisation technique [31]. The cycle ends when the number of training times is reached, and the lowest loss error will be the output.

### 4.2. Using the LSTM Model to Predict the State of Cattle

It is critical to determine the input, output, and time series before using the constructed LSTM model for cattle state prediction. The cattle’s state must be presented as the output, and the number of the independent variable hours must be seen as a time series, according to the characteristics of the data sets. As a result, determining input variables is a challenging aspect of this approach. Because the output variable must be data with periodic changes, the input must be a known fixed periodic function. Time series as a fixed periodic function can be used as input. To be more specific, given that the state cycle of cattle is one day, it is appropriate to determine the input variable as the number of hours on the clock each day. The input and output variables, as well as the time series, for the resting time of Brahman Female’s cattle are as follows:Input: The number of hours on the clock each day (24 h).Output: The resting time during this hour (e.g., The resting time at 7:00 means that the resting time during one hour from 7:00 to 7:59).Time series *t*: The sequence number of this hour (e.g., 0:00 a.m. on the first day is the first hour, and *t* is 1. So on, 0:00am on the second day is the 25th h, and *t* is 25) [30,32].
Training:Both the input and output data are periodicities. The distinction is that the input in this cycle has a set value and trend, whereas the output in each cycle has a varied value. For example, the input is 0 at 0:00 a.m. on Day 17th and 0:00 a.m. on Day 24th, as shown by the two red lines in Figure 11, but the output is different. In other words, the same input might result in multiple outcomes regardless of time. Although the input is the same, the input’s matching time series is not. As a result, when a single input correlates to numerous outputs in a time series, the LSTM model can successfully handle the problem.Testing and prediction:In total, 90% of the data is used for training, and 10% for prediction and testing. For example, the input data sets for training are $input_{t_{1}}$ through $input_{t_{90}}$, while the data sets for testing are $input_{t_{91}}$ through $input_{t_{100}}$. The training outcomes are depicted in Figure 12.

The predict and actual results are similarly shown in Figure 12. This means the digital twin model for individual cattle is basically established. The training loss reduces during the training process, showing that the model is converged and practical in Figure 13. However, the prediction results’ error is relatively significant, which indicates further requirements of the parameter optimization in the model.

**Figure 11 sensors-22-07118-f011:**
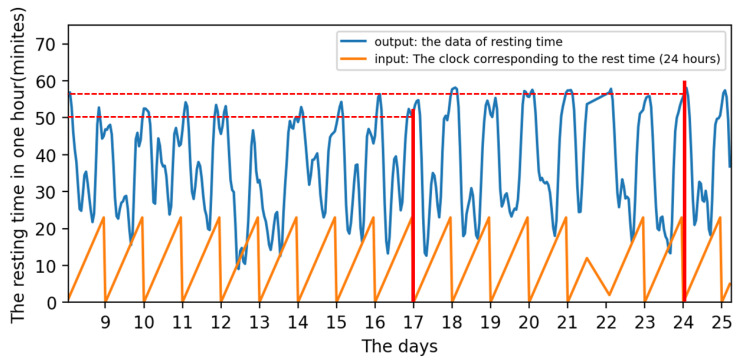
The input and output based on the LSTM model.

**Figure 12 sensors-22-07118-f012:**
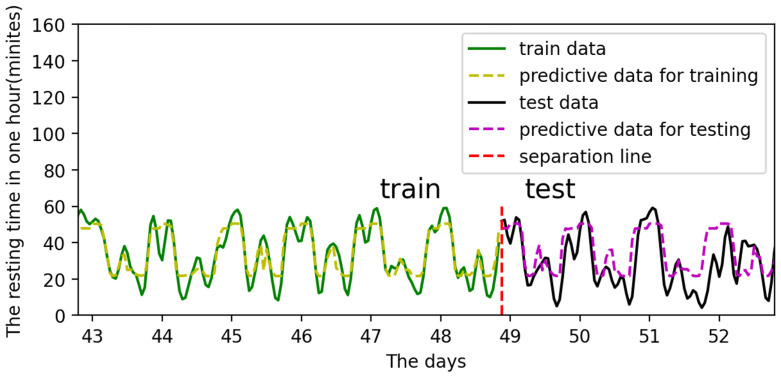
The predictive results after training and testing.

**Figure 13 sensors-22-07118-f013:**
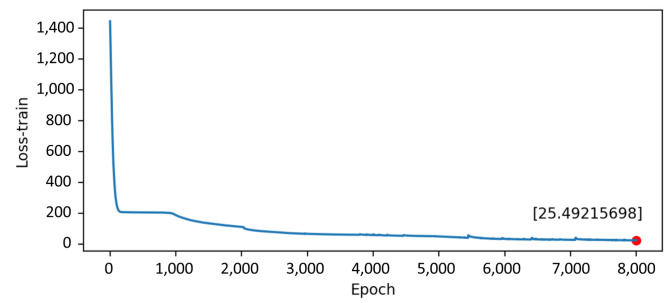
The training loss.

### 4.3. Parameter Optimization

For optimization and comparison purposes, the number of hidden units, LSTM layers, the batch size, and the epoch size were all modified [29,30].
Hidden units size: 4, 8, 16, 32, 64, 128, 256.The number of LSTM layers: 1, 2, 3, 4, 5, 6, 7.The batch size: 3, 6, 12, 24, 48, 96.The epoch size: 100, 500, 1000, 2000, 5000, 10,000, 20,000.
Selection of the number of LSTM layersThe number of hidden units is 16, the batch size is 24, and the epoch size is 2000, all of which are randomly chosen. Only the number of layers in the LSTM is modified with the other parameters fixed: 1, 2, 3, 4, 5, 6, 7. The box diagram for the mean square deviation in the model learning process is shown in Figure 14.The top line and bottom line represent the edge’s maximum and minimum values, respectively. The upper quartile is represented by the box’s upper edge, while the box’s lower edge represents the lower quartile. The orange line represents the median. Comparing the seven box charts, increasing the number of layers has a minor impact on the mean square error of model training [33].When the number of layers is 5, 6 and 7, the error of the LSTM model will be stabilized to a fixed value immediately after a short training. As shown in Figure 14, the box plot has many outliers (that is, large outliers, black circles in the figure), and the median, upper quartile, and lower quartile overlap. However, in terms of model performance, using more LSTM layers, the running speed will be slower and it becomes more complex, and the result of the model operation is affected [34,35]. The loss error of the test set is positively correlated with that of the training set, and it is the smallest when the number of layers is 2. As a result, two layers of LSTM are best for this model.Selection of the hidden units sizeTo determine the size of the hidden units, we keep the batch size and epoch size unchanged and run the LSTM model with different hidden units size, i.e., 4, 8, 16, 32, 64, 128, 256. The box diagram of the mean square is shown in Figure 15. In terms of error size and ultimate training effect, the choice of 128 hidden units is the best for training the data, with the majority of the mean square error values falling below 25, and the loss error of the test set is the smallest.Selection of the batch sizeThe batch size, which can be 3, 6, 12, 24, 48, or 96, is altered when using two layers of LSTM with 128 hidden units. The box diagram is shown in Figure 16. The batch size refers to the number of samples fed into the model at once and divides the original data set into batch size data sets for independent training. This method helps to speed up training while also consuming less memory [36]. To some extent, batch size training can help to prevent the problem of overfitting [37]. As a result, when building the model, an acceptable batch size should be chosen. When the batch size is 24, the minimum value of the produced mean square deviation data set is the smallest in terms of minimum value and median, as well as the test error value.Selection of the epoch sizeSelect two layers of LSTM with 128 hidden units and the batch size is 24, but the epoch size can be any of 100, 500, 1000, 2000, 5000, 10,000, or 20,000. Figure 17 shows a box diagram for the mean square deviation in the model learning process.The epoch size is the number of times the learning algorithm works in the entire training data set. An epoch means that each sample in the training data set has the opportunity to update internal model parameters [38]. In theory, the more training sessions there are, the better the fit and the lower the error. In practice, however, overfitting occurs when the epoch size exceeds a specific threshold, causing the training outcomes to deteriorate [39]. The epoch size of 100, 500, 1000, 2000, 5000, 10,000, and 20,000 is chosen in Figure 17. The inaccuracy rapidly decreases and approaches zero as the epoch size increases from 100 to 10,000. When the epoch size increases to 20,000, the error is still tiny, but it is greater than when the epoch size is 10,000, indicating an overfitting occurrence. Therefore, the model with a 10,000 epoch size has the best effect.

Figure 18 shows the training and prediction outcomes after optimizing model parameters, while Figure 19 shows the loss value after optimizing parameters. The best parameters for the LSTM model are shown in Table 4. The LSTM model has a good prediction of the resting state of cattle, which largely adheres to the periodic changes in cattle state and has a modest error. Therefore, the digital twin model for cattle has been established and optimized.

**Figure 14 sensors-22-07118-f014:**
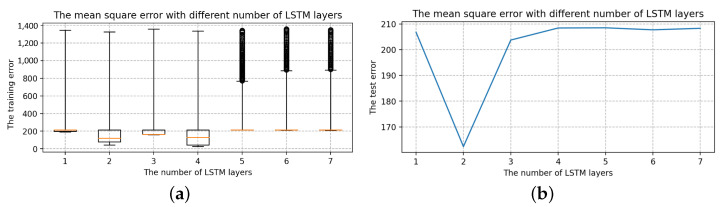
The mean square error with different numbers of LSTM layers. (**a**) The training error with different number of LSTM layers. (**b**) The test error with different number of LSTM layers.

**Figure 15 sensors-22-07118-f015:**
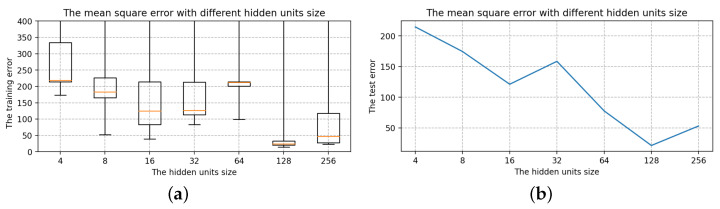
The mean square error with different hidden units size. (**a**) The training error with different hidden units size. (**b**) The test error with different hidden units size.

**Figure 16 sensors-22-07118-f016:**
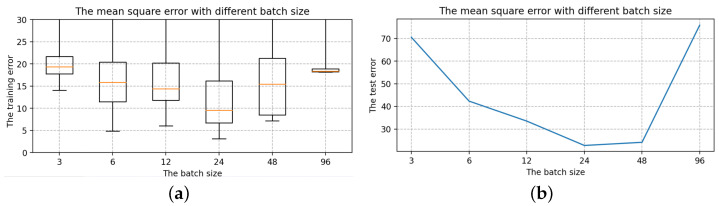
The mean square error with different batch sizes. (**a**) The training error with different batch sizes. (**b**) The test error with different batch sizes.

**Figure 17 sensors-22-07118-f017:**
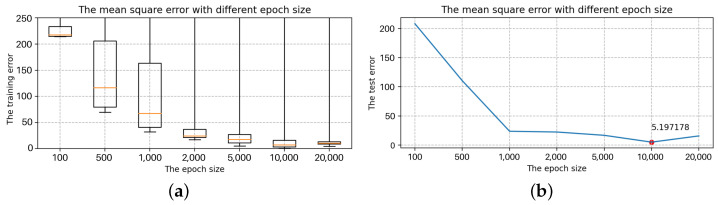
The mean square error with different epoch sizes. (**a**) The training error with different epoch sizes. (**b**) The test error with different epoch size.

**Figure 18 sensors-22-07118-f018:**
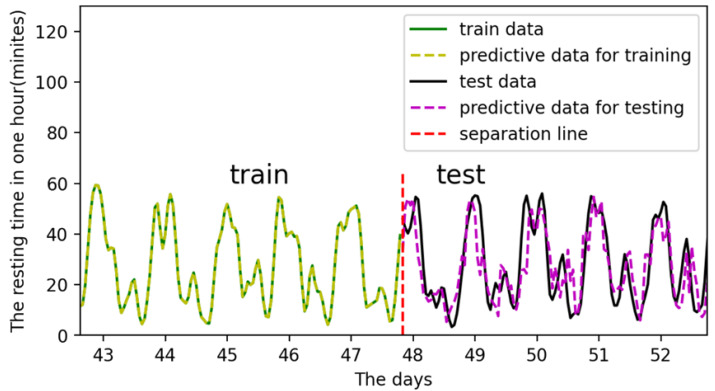
The training and prediction after optimizing model parameters.

**Figure 19 sensors-22-07118-f019:**
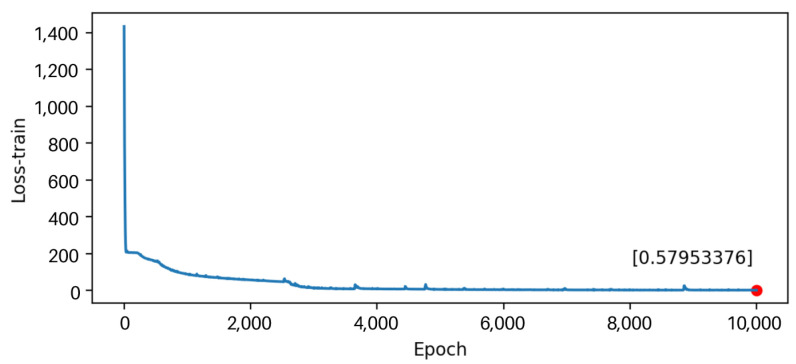
The training loss after optimizing model parameters.

## 5. Results and Analysis

Figure 20 depict the LSTM model’s training and prediction on different sexes, breeds, and states, respectively. This shows the applicability of this model, which can be used to predict various states of different cattle.

The trend of the results predicted by this LSTM model is nearly identical to the actual data. The model for Brahman males performs relatively poorly, which can be attributed to their relatively random rest state, poor cycle regularity, and other external environmental factors. It is possible that increasing the size of the data collection may result in improved predictions. Overall, the LSTM-based model for the cattle state cycle is accurate and effective, and it can accurately predict the dynamic trend of the next cattle state cycle.

In this way, the digital twin model can effectively predict the future time budget of cattle, which is conducive to efficient cattle breeding. Predict the future behaviour of cattle in advance so that appropriate preventive measures can be prepared.

## 6. Conclusions

The construction of a smart digital twin model of the state of cattle is primarily achieved in this work. It is primarily built on a farm IoT system to collect the state data of cattle under various combined treatments, with data cleaning and calculating. The average data of 24 h are fitted, and the data of the whole sampling period are de-noised. In addition, a deep learning-based LSTM model for cattle state dynamics is developed using the data after noise reduction, and the model can predict the state change of cattle in the next cycle. The model’s accuracy and effectiveness are demonstrated when the prediction results are compared to the actual results. After optimization, the loss error of the training set is reduced to about 0.580, and the loss error of the prediction set is about 5.197. Using this digital twin model, the future time budget of cattle can be predicted quickly and accurately.

This model has certain limits as well, it requires a large quantity of data to learn, and a little amount of data will cause the model to be inaccurate. Furthermore, encapsulating the entire research into one system is a critical step toward commercializing digital twins in the future. In addition, estimating the time budget of cattle in advance necessitates human prediction of cow health conditions. Fully automated cow feeding and real-time monitoring of cattle condition and health are desirable in the future.

## Figures and Tables

**Figure 1 sensors-22-07118-f001:**
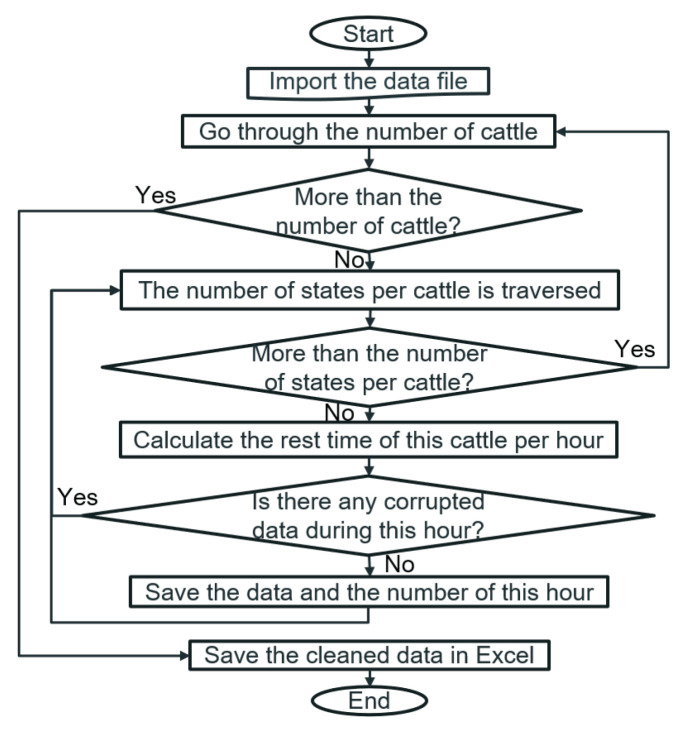
Process flow chart of data cleaning.

**Figure 2 sensors-22-07118-f002:**
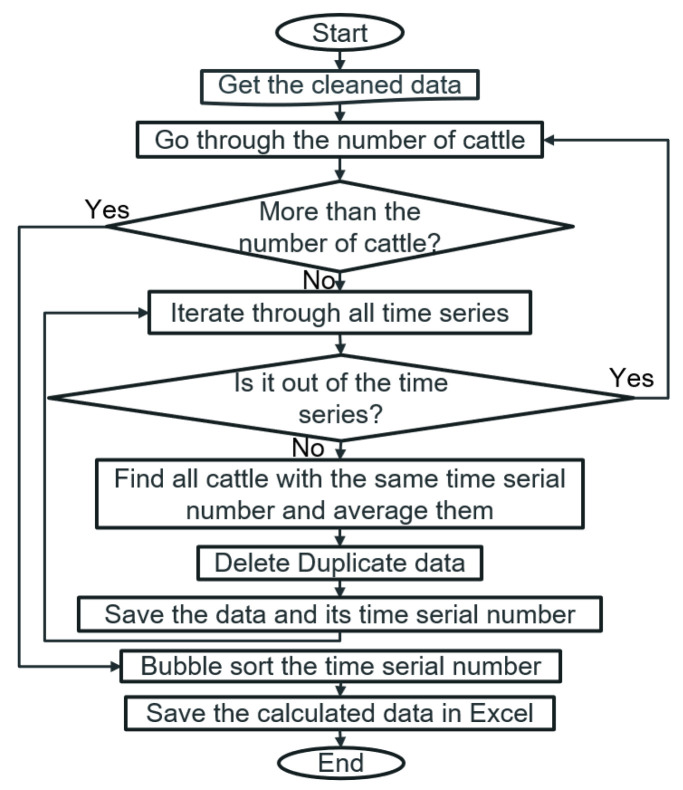
Process flow chart of average state times for multiple cattle.

**Figure 3 sensors-22-07118-f003:**
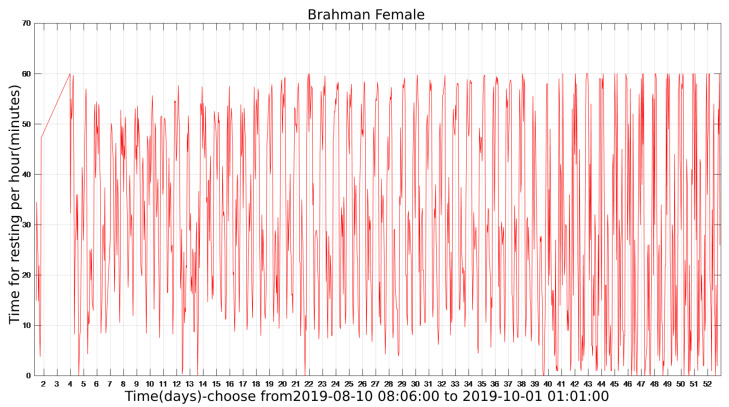
The resting time of Brahman Female during the whole sample period.

**Figure 4 sensors-22-07118-f004:**
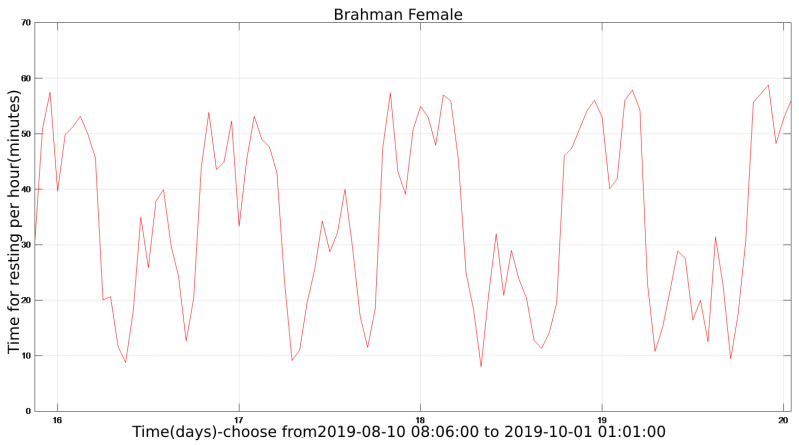
The enlarge vision shown 4 days.

**Figure 5 sensors-22-07118-f005:**
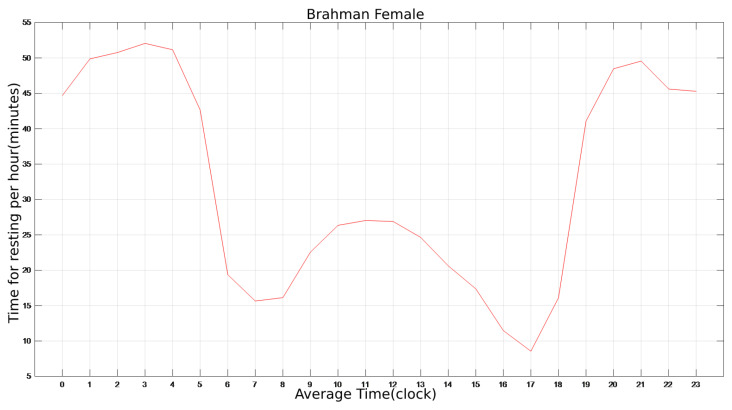
The average of one resting period for cattle(i.e., 24 h a day).

**Figure 6 sensors-22-07118-f006:**
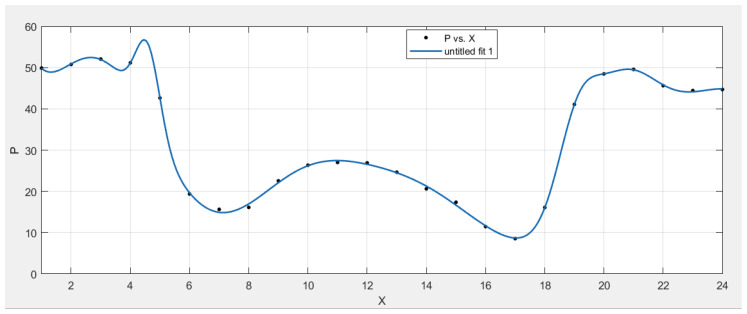
The Gaussian Fitting for one average period.

**Figure 7 sensors-22-07118-f007:**
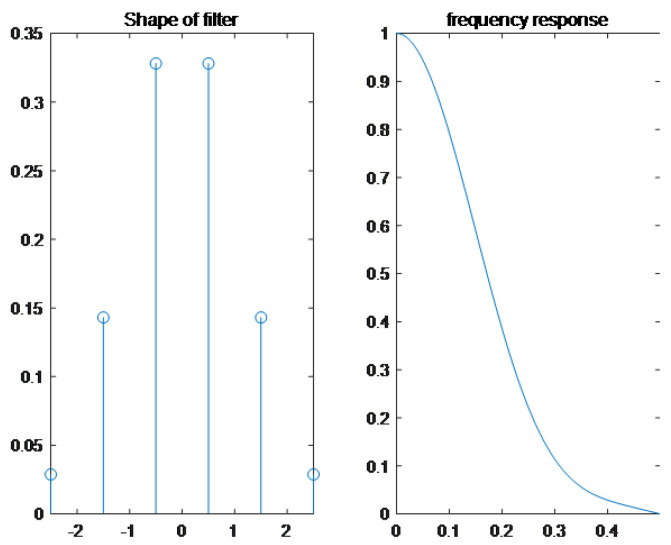
The shape of the FIR filter and the frequency response.

**Figure 8 sensors-22-07118-f008:**
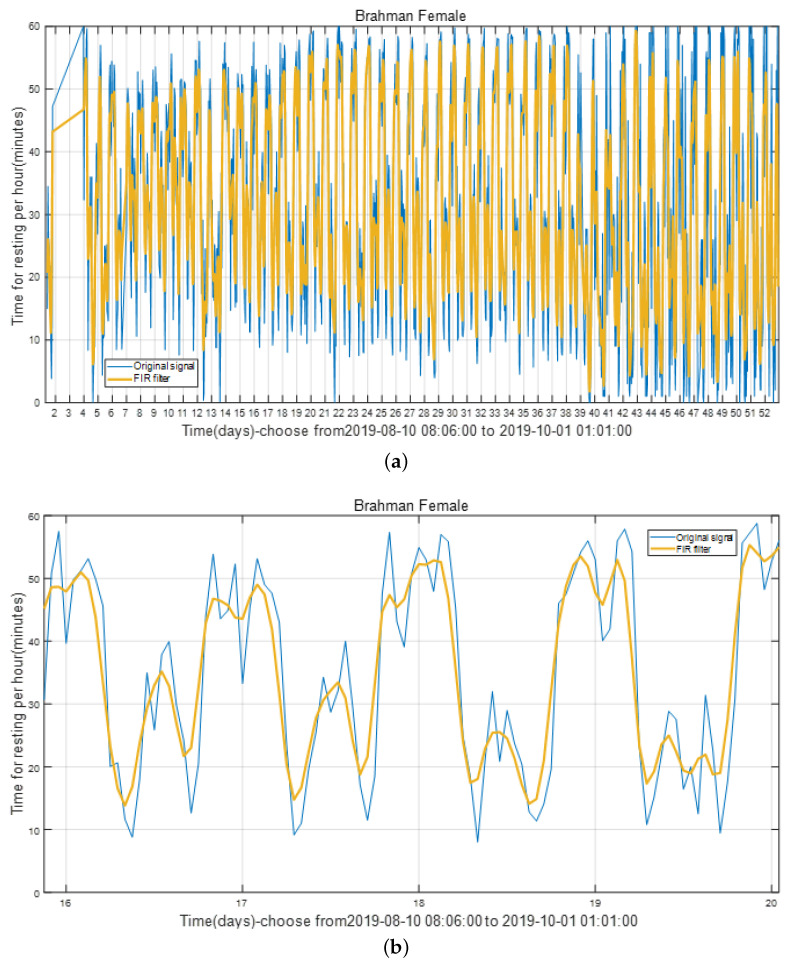
The resting time of cattle during the whole period after using the FIR filter. (**a**) The whole sample period. (**b**) The enlarge vision shown 4 days.

**Figure 9 sensors-22-07118-f009:**
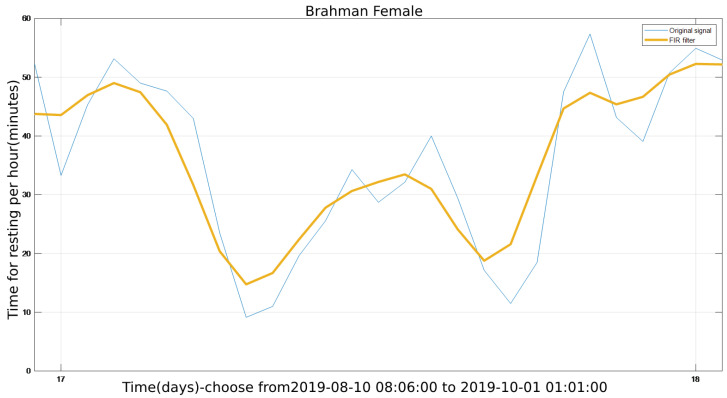
The single resting cycle after the FIR filter.

**Figure 10 sensors-22-07118-f010:**
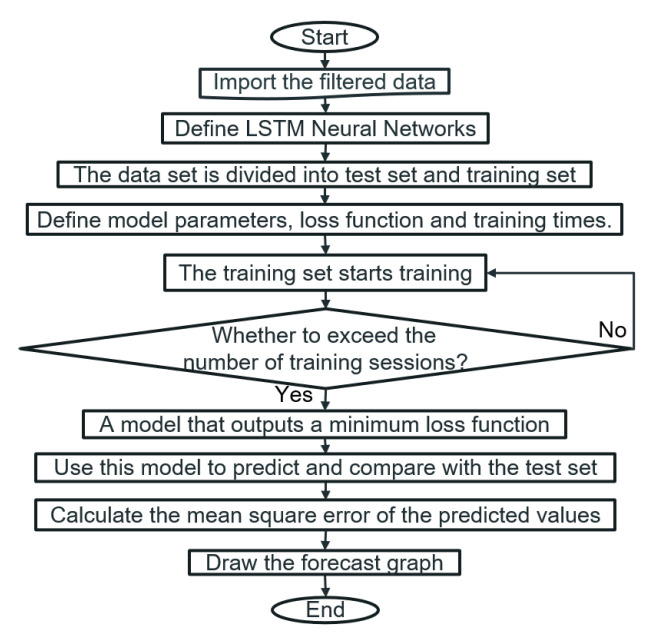
The code process of building the LSTM model.

**Figure 20 sensors-22-07118-f020:**
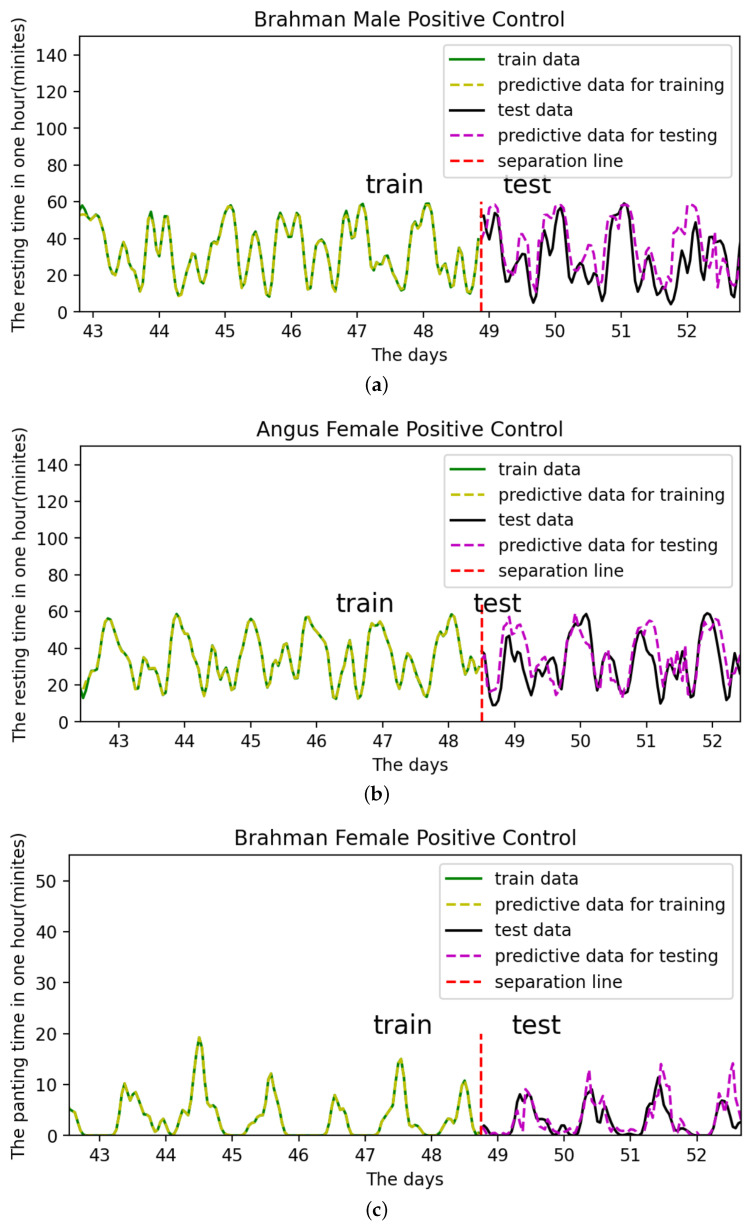
Applicability of the model. (**a**) Brahman Male cattle (Rest). (**b**) Angus Female cattle (Rest). (**c**) Brahman Female cattle (Pant).

**Table 1 sensors-22-07118-t001:** The explanation of the state.

State	Description
Rest	Standing still, lying, and transition between these 2 events, while lying, allowed to do any kind of movement with head/neck/legs (e.g., tongue rolling).
Rumination	Rhythmic circular/side-to-side movements of the jaw not associated with eating or medium activity, interrupted by brief (<5 s) pauses during the time that bolus is swallowed, followed by a continuation of rhythmic jaw movements.
Panting (Heavy Breathing)	Fast and shallow movement of thorax visible when looking animal from side, along with forward heaving movement of body while breathing. May or may not have open mouth, salivation, and/or extended tongue.
High Activity	Includes any combination of running, mounting, head-butting, repetitive head-weaving/tossing, leaping, buck-kicking, rearing and head tossing.
Eating	Muzzle/tongue physically contacts and manipulates feed, often but not always followed by visible chewing.
Grazing	Eating (see above definition) growing grass and pasture, while either standing in place or moving at slow, even or uneven pace between patches.
Walking	Slow movement, limb movement, except running.
Medium Activity	Any activity other than the above states.

**Table 2 sensors-22-07118-t002:** The number of cattle of different breeds and genders.

Category	Number
Angus Female	13
Angus Male	14
Brahman Female	14
Brahman Male	5
Brangus Female	10
Brangus Male	0
Charolais Female	3
Charolais Male	1
crossbred Female	38
crossbred Male	0
Total number	98

**Table 3 sensors-22-07118-t003:** The results of different fitting methods.

Fitting Method	The Best Number of Items	Variance
Gaussian Fitting	8	3.0037
Sum of sine	8	20.1288
Polynomial	9	245.3264
Fourier	8	25.4590

**Table 4 sensors-22-07118-t004:** The best parameters for the LSTM model.

Hidden Neurons	128
Batch size	24
Epoch size	10,000
LSTM layers	2
Loss-train	0.57953376
Loss-test	5.197178

## Data Availability

Not applicable.

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
