# Peer review of "AI Based Digital Twin Model for Cattle Caring"

_sensors, 2022, doi:10.3390/s22197118_

Round 1
Reviewer 1 Report
The manuscript titled ' AI Based Digital Twin Model for Cattle Caring
' has been organized well. The language of the manuscript is clear. The required literature summary and material and methods have been presented appropriately. I found the paper to be somewhat interesting. However, the below issues should be addressed if the authors would like to pursue its publication.
- specify the accuracy in abstract.
-
In the introduction, the motivation of the paper needs to be articulated far more clearly.
-
Furthermore, where are the limitations of your study? Clarifying the limitations of a study allows the readers to understand better under which conditions the results should be interpreted.
- Clearly specify the contribution in introduction section.
- Very limited literature review is presented, add LR from more recent papers.
- Fig.1 The digital twin model of the cattle, what does it mean?
- Fig 2 is not readable. Modify.
- Provide statistical summary of data set.
- Labels of fig 5 are not readable.
- Why Gaussian fitting is preferred? justify.
- No need to separate conclusion in two separate sections.
Reviewer 2 Report
The review is added as PDF files. (vgl. review.pdf) Please cover all concerns.

Round 2
Reviewer 2 Report
Dear authors, I reviewed your updated version of the paper and see that you corrected the main concern made by me on your initial submission. With the parts about the cattle's pain deleted, the soundness of the work is completely given. The other parts criticized, like the missing references to existing literature or the quality of the plots, have been refined as well.
Besides a final check of the spelling, the paper should be ready to be published.
All the Best,
reviewer 2